# New Approach to Non-Invasive Tumor Model Monitoring via Self-Assemble Iron Containing Protein Nanocompartments

**DOI:** 10.3390/nano12101657

**Published:** 2022-05-12

**Authors:** Anna N. Gabashvili, Maria V. Efremova, Stepan S. Vodopyanov, Nelly S. Chmelyuk, Vera V. Oda, Viktoria A. Sarkisova, Maria K. Leonova, Alevtina S. Semkina, Anna V. Ivanova, Maxim A. Abakumov

**Affiliations:** 1Laboratory “Biomedical Nanomaterials”, National University of Science and Technology “MISiS”, Leninskiy Prospect, 4, 119049 Moscow, Russia; gabashvili.anna@gmail.com (A.N.G.); stepan.vodopianov@yandex.ru (S.S.V.); nellichmelyuk@yandex.ru (N.S.C.); oda-vera@mail.ru (V.V.O.); leonovamasha19@gmail.com (M.K.L.); super.fosforit@yandex.ru (A.V.I.); 2Transplantation Immunology Laboratory, Biomedical Technology Department, National Medical Research Center for Hematology, Novy Zykovsky Drive, 4A, 125167 Moscow, Russia; 3Department of Chemistry and TUM School of Medicine, Technical University of Munich, Ismaninger Str. 22, 81675 Munich, Germany; m.efremova@tue.nl; 4Institute for Synthetic Biomedicine, Helmholtz Zentrum München GmbH, Ingolstaedter Landstr. 1, 85764 Neuherberg, Germany; 5Department of Applied Physics, Eindhoven University of Technology, Cascade P.O. Box 513, 5600 MB Eindhoven, The Netherlands; 6Biology Faculty, Lomonosov Moscow State University, Leninskiy Gory, 119234 Moscow, Russia; alice-lyddell@yandex.ru; 7Department of Medical Nanobiotechnology, Pirogov Russian National Research Medical University, Ostrovityanova St., 1, 117997 Moscow, Russia; alevtina.semkina@gmail.com; 8Cell Proliferation Laboratory, Engelhardt Institute of Molecular Biology, Vavilova Street, 32, 119991 Moscow, Russia; 9Department of Basic and Applied Neurobiology, Serbsky National Medical Research Center for Psychiatry and Narcology, Kropotkinskiy Per. 23, 119991 Moscow, Russia

**Keywords:** encapsulins, magnetic resonance imaging, fluorescence, cell tracking

## Abstract

According to the World Health Organization, breast cancer is the most common oncological disease worldwide. There are multiple animal models for different types of breast carcinoma, allowing the research of tumor growth, metastasis, and angiogenesis. When studying these processes, it is crucial to visualize cancer cells for a prolonged time via a non-invasive method, for example, magnetic resonance imaging (MRI). In this study, we establish a new genetically encoded material based on *Quasibacillus thermotolerans* (*Q.thermotolerans, Qt*) encapsulin, stably expressed in mouse 4T1 breast carcinoma cells. The label consists of a protein shell containing an enzyme called ferroxidase. When adding Fe^2+^, a ferroxidase oxidizes Fe^2+^ to Fe^3+^, followed by iron oxide nanoparticles formation. Additionally, genes encoding mZip14 metal transporter, enhancing the iron transport, were inserted into the cells via lentiviral transduction. The expression of transgenic sequences does not affect cell viability, and the presence of magnetic nanoparticles formed inside encapsulins results in an increase in T2 relaxivity.

## 1. Introduction

Usually, tumor cells’ in vivo visualization is carried out using optical methods, as well as MRI and positron emission tomography (PET) methods. For example, it is possible to perform two-color fluorescent imaging using tumor cells expressing red fluorescence protein (RFP) transplanted into transgenic mice expressing green fluorescence protein (GFP) [1]. In addition, there are two-color fluorescent cells with RFP expressed in the cytoplasm and GFP expressed in the nucleus [2]. Another example of fluorescent labels is quantum dots (QDs), which are inorganic fluorescent nanoparticles with great optical properties such as high quantum yield, high molar extinction coefficients, high resistance to photo bleaching and chemical decomposition, and restricted emission spectra, compared to organic dyes [3]. A common disadvantage of fluorescent labels, both exogenous labels and genetically encoded proteins, is a small penetration depth (about 2 mm), which makes their use only possible for surface tissues in small animals, for example, for subcutaneous tumor imaging in mice. In addition, a significant disadvantage of exogenous fluorescent labels is the decrease in the intensity of the fluorescent signal associated with cell proliferation.

Superparamagnetic iron oxide (SPIO) nanoparticles [4,5] are often considered for MRI tumor cells tracking. One of the drawbacks of SPIO nanoparticles is the negative contrast they produce that is often difficult to interpret. An alternative is to use gadolinium-based agents that give positive contrast on T2-weighted MRI scans. For example, in [6], rhodamine-conjugated gadolinium nanoparticles were utilized to label and track breast carcinoma cells in vivo. Cells labeled by such nanoparticles can be visualized using both MRI and optical imaging; however, the same problem as when using fluorescent labels remains—the intensity of the MR signal decreases over time due to cell proliferation. In addition, nanoparticles can be ejected by tumor cells and internalized by phagocytic cells in the tumor microenvironment, which degrades imaging accuracy.

Finally, more than a decade ago, PET was introduced to visualize malignant cells using the labels with various half-lives such as ^18^F-FDG or ^64^Cu-PTSM [7]. Their main disadvantages are toxicity and short half-life, which do not allow the long-term monitoring of tumor cells in vivo.

In light of the above, developing a stable and non-toxic label that allows the non-invasive tracking of tumor cells remains highly relevant. Encapsulins, protein nanocompartments with a high molecular weight, homologous in structure to viral capsids (icosahedral shells consisting of protomer proteins), were first discovered in 1994 in *Brevibacterium linens* [8]. Later, it was found that encapsulins occur in many other bacterial strains and archaea [9,10,11]. *Q. thermotolerans* encapsulin can accumulate more than 30,000 iron atoms inside its shell, which is an order of magnitude more than the amount of iron accumulated in ferritins. The diameter of the *Q.thermotolerans* encapsulin shell is large (42 nm) and consists of 240 protomers (32.2 kDa each). The accumulation of iron inside the encapsulin shell results from a catalytic activity of IMEF (Iron-Mineralizing Encapsulin-Associated Firmicute) cargo protein. It is assumed that, in wild-type *Q. thermotolerans*, encapsulins protect cells against oxidative stress by depositing iron inside the nanocompartments, thereby reducing the amount of free iron in the cytoplasm [12]. In this paper, we present *Q. thermotolerans* encapsulin-based genetic label, which is stably expressed in 4T1 mouse carcinoma cells.

To obtain a 4T1 cell line with a stable expression of *Q. thermotolerans* encapsulins genes (4T1-Qt), lentiviral transduction was performed using two viral vectors that carry genes encoding encapsulin shell and cargo protein (QtEnc^FLAG^-QtIMEF), as well as genes encoding the divalent metal transporter (mZip14). We used ferrous ammonium sulfate (FAS) as a source of Fe^2+^. The entire system works as follows: Fe^2+^ ions from FAS are transported into cells via mZip14 and enter the encapsulin nanocompartments. There, under the action of IMEF, Fe^2+^ is oxidized, resulting in iron oxide nanoparticles, which further allow 4T1-Qt cell tracking by MRI.

## 2. Materials and Methods

### 2.1. Cell Line

4T1 mammary mouse carcinoma cells were cultured in RPMI 1640 media (Gibco, Waltham, MA, USA) supplemented with antibiotics (100 U/mL penicillin, 100 mg/mL streptomycin, Gibco), GlutaMax Supplement (2 mM, Gibco), and 10% fetal bovine serum (HyClone, Cytiva, Washington, DC, USA). The cells were cultured under standard conditions (37 °C and 5% CO_2_) in T-75 cultural flasks (Corning, New York, NY, USA) and used between 5 and 13 passages. Upon reaching high confluence, the cells were subcultured at a ratio of 1:4–1:6 following the standard trypsinization method.

### 2.2. Lentivirus Production and Lentiviral Transduction of Cells

HEK293T packaging cells were seeded in 6-well tissue culture plates (3.0 × 10^6^ cells per plate) and cultured in complete DMEM medium (Gibco) at 37 °C and 5% CO_2_. After 24 h of cultivation, lentiviral packaging plasmids (pRSV-Rev, pMDLg/pRRE, pCMV-VSV-G), plasmid-carrying encapsulin genes (pLCMV QtEnc^FLAG^-QtIMEF), and plasmid encoding genes of iron transporter (pLCMV mZip14) were added to the cells in Opti-MEM medium. Individual lentiviral vector was constructed for QtEnc^FLAG^-QtIMEF and mZip14. Then, 24 h after transfection, the medium was aspirated and replaced with DMEM supplemented with 2–5% FBS and antibiotics. Viruses were harvested 48 and 72 h post-transfection, loaded onto a 20 mL syringe, and filtered through a 0.45 µm syringe filter (Merck). Transduction 4T1 carcinoma cells with the lentiviral vectors was performed according to standard protocol in DMEM growth medium supplemented with 10% heat-inactivated FBS and polybrene (8 μg/mL, Sigma-Aldrich, Darmstadt, Germany). Lentiviruses were added to give a MOI of 4 for each virus; 48 h after transduction, the selection was started using puromycin (Thermo Fisher Scientific, Waltham, MA, USA) at a concentration of 5 µg/mL, and the medium with puromycin was changed to a new one every other day. 

### 2.3. Reverse Transcription Polymerase Chain Reaction (RT-PCR)

The cells obtained after lentiviral infection were analyzed by reverse transcription polymerase chain reaction. Total RNA was extracted by Extract RNA reagent (Evrogen, Moscow, Russia) according to the manufacturer’s protocol. RNA concentrations and quality were assessed by spectrophotometry. Then, cDNA was synthesized with Invitrogen SuperScript III Reverse Transcriptase (Thermo Fisher, Waltham, MA, USA) and oligo-DT and random primers. The cDNA and negative control (RNA with no reverse transcriptase added) were used for classical PCR with Taq polymerase (Fermentas, Waltham, MA, USA). The products of PCR were separated in a 1% agarose gel electrophoresis. The amplified fragments were identified by their length.

### 2.4. Western Blot Analysis

Western blot analysis was carried out as described earlier [13]. Briefly, 4T1-Qt cells were lysed using RIPA buffer, and the resulting lysate was precipitated (15 min, 14,000× *g*). Sample buffer 5× was added to different amounts of cell lysate, heated at 95 °C, then cooled on ice. Samples were loaded onto a gel and electrophoresed (80 V for 25 min and 100 V for 1.5 h); then, the gel was transferred into a transfer buffer. The nitrocellulose membrane was activated and placed over the gel. The transfer was carried out in a chamber filled with transfer buffer for 1 h at 100 V. Then, the membrane was washed three times to remove transfer buffer residues in PBST. To prevent nonspecific binding, the membrane was incubated in a PBST solution with 5% non-fat milk for 2 h and washed again. The membrane was incubated with anti-DYKDDDDK Tag antibodies (1:1000, BioLegend, San Diego, CA, USA) for 2 h, followed by washing three times. After that, alkaline horseradish peroxidase conjugated secondary antibodies (1:1000, goat anti-mouse IgG, Santa Cruz Biotechnology,) were added. Clarity Max Western ECL Substrate kit (BioRad, Hercules, CA, USA) was used to reveal the result. The results were registered with the ChemidocMP Imaging system (BioRad, Hercules, CA, USA).

### 2.5. Immunofluorescence Staining

For direct immunofluorescence analysis, 4T1 and 4T1-Qt cells were seeded on a 35 mm µ-Dish with a polymer coverslip bottom for high-end microscopy (Ibidi, Grafelfing, Germany) in the amount of 5 × 10^4^ cells/dish, cultured for 24 h in standard conditions, fixed by 4% formaldehyde in 1 × PBS (Sigma-Aldrich, Darmstadt, Germany), and stained by Monoclonal Antibody to DYKDDDDK Tag (L5), Alexa Fluor 647 (1:250, BioLegend, San Diego, CA, USA), according to the manufacturer’s instructions. Nuclei were counterstained with DAPI (1:500, Sigma Aldrich, Darmstadt, Germany).

### 2.6. Laser Scanning Confocal Microscopy

Fluorescence confocal micrographs were captured with the Nikon Eclipse Ti2 microscope (Minato, Tokyo, Japan) equipped with ThorLabs laser (Newton, NJ, USA) and scanning systems, Nikon Apo 25×/1.10 water immersion objective lens (Minato, Tokyo, Japan), Nikon Plan Apo 10×/0.45 objective lens (Minato, Tokyo, Japan), and 405 and 642 lasers. Scanning was performed using the ThorImageLS software (Newton, NJ, USA), ImageJ2 FiJi was used to process the images.

### 2.7. MTS-Assay

Cytotoxicity FAS for 4T1 and 4T1-Qt cells were performed via CellTiter 96 AQueous One Solution Cell Proliferation Assay (Promega, Madison, Wisconsin, USA) according to manufacturer’s protocol. 4T1 and 4T1-Qt cells were seeded at 8 × 10^3^ cells/well in a 96-well culture plate in 100 μL of the medium per well. After 24 h of incubation, FAS (Sigma-Aldrich, Darmstadt, Germany) at different concentrations (4 mM, 2 mM, 1 mM, 0.5 mM, 0.25 mM, and 0.13 mM) was added to the cells. After 24 h incubation, the cells were washed with PBS, and a fresh growth medium with MTS reagent was added in each well. Non-treated cells by FAS were used as a positive control. The cells were incubated with MTS reagent for 4 h at 37 °C and 5% CO_2_ in the humid atmosphere. The assay was conducted in three replicates. Optical density was measured using a Multiscan GO plate reader (Thermo Scientific, Waltham, MA, USA), λ = 490 nm. 

Cell viability was calculated as:Cell viability (%) = (A_s_ − A_b_)/(A_c_ − A_b_) × 100(1)
where A_s_—mean optical density in sample wells, A_b_—mean optical density in blank wells, A_c_—mean optical density in positive control wells. 

### 2.8. Magnetic-Activated Cell Sorting (MACS)

The magnetic sorting of 4T1 and 4T1-Qt cells after 24 h of incubation with 2 mM FAS was performed using a magnetic separation kit (Miltenyi Biotech, Bergisch Gladbach, North Rhine-Westphalia, Germany). The cells were thoroughly washed from FAS with PBS and then removed from the plastic by trypsinization, precipitated (500 g, 5 min), and resuspended in 1.5 mL of PBS containing 2% FBS, and the number of cells was counted using an automatic cell counter. MS columns were placed in the magnetic field of an OctoMACS Separator and equilibrated with 0.5 mL PBS per column (Gibco, Waltham, MA, USA). The cell suspension was applied to magnetic columns, and free-passing cells were collected in a 15 mL tube. The columns were washed 3 times with 1 mL of PBS containing 2 % FBS, removed from the separator’s magnetic field, and placed in a new 15 mL tube. Then, 1 mL of PBS with 2% FBS was added to the column, and the cells retained in the column were eluted using a plunger. Next, the cells were pelleted (500 g, 5 min), resuspended in a complete growth medium, and counted. The isolated cells were further cultured under standard conditions.

### 2.9. Prussian Blue Staining

4T1 and 4T1-Qt cells were seeded at 2 × 10^5^ cells/dish in 35 mm Petri dishes and cultured for 24 h. FAS (4 mM, 2 mM, 1 mM, 0.5 mM, 0.25 mM, and 0.13 mM) was added to the cells, and they were incubated with FAS for 24 h. Afterward, the cells were thoroughly washed with PBS, fixed by 4% formaldehyde in PBS, and then stained using Iron Stain Kit (Sigma-Aldrich), according to the manufacturer’s instructions. Images of 4T1 and 4T1-Qt cells were taken with an inverted microscope Primo Vert (Zeiss, Oberkochen, Germany).

### 2.10. Transmission Electron Microscopy (TEM)

4T1-Qt cells were seeded in µ-Slide 8 Well (Ibidi, Grafelfing, Germany) at a concentration of 4 × 10^4^ cells/well and cultured in a growth medium (composition described above) supplemented with 2 mM FAS for 24 h. Afterward, the cells were thoroughly washed with PBS and fixed with 2% paraformaldehyde and 2.5% glutaraldehyde (Sigma-Aldrich, Darmstadt, Germany) in PBS, pH 7.4. Then, the cells were postfixed with 1% osmium tetroxide solution and dehydrated in ethanol of increasing (50%, 70%, 80%, and 95%) concentration. Finally, the cell sample was embedded in an Epoxy resin using an Epoxy embedding medium kit (Sigma Aldrich) according to the manufacturer’s protocol. Ultrathin (70 nm) cell sections were obtained with an EM UC6 ultramicrotome (Leica, Wetzlar, Germany). Imaging was performed with a JEOL JEM 1400 electron microscope (Akishima, Tokyo, Japan).

### 2.11. Measurement of Intracellular Iron Content by Atomic Emission Spectroscopy (AES)

4T1-Q and 4T1 (control) cells were cultured in 6-well plates (5 × 10^5^ cells/well). To investigate the iron accumulation in cells, the growth medium was supplemented with 2 mM, 1 mM, 0.5 mM, and 0.25 mM FAS for 24 h. Then, the cells were thoroughly washed with PBS, detached with 0.25% trypsin solution, precipitated by centrifugation, and counted. For each cell line, 1.6 × 10^6^ cells were dissolved in 70 μL of concentrated nitric acid for 2 h at 60 °C. Iron concentration was determined using an Agilent 4200 MP-AES atomic emission spectrometer (Santa Clara, CA, USA).

### 2.12. MRI

For T2 relaxometry measurement, 4T1 and 4T1-Qt cells were incubated with 2 mM and 1 mM FAS for 24 h. Afterward, cells were washed 3 times with DPBS, detached with TripLE, and centrifuged at 500× *g* for 5 min. The pellets (5 × 10^6^ cells each) were resuspended in 200 μL DPBS and transferred to 500 μL PCR tubes. Cells were then spun down at 500× *g* for 2 min and used for MRI. MRI images were acquired on ClinScan 7T system (Bruker Biospin, Billerica, MA, USA) in Spin Echo sequence with the following parameters: TR = 10,000, slice thickness 1.2 mm, FoV 84 × 120, base resolution 448 × 640, TE 8, 16, 24,…, 256.

### 2.13. Animals and Tumor Model

All manipulations with experimental animals were approved by the local Ethical Committee of the Pirogov Russian National Research Medical University. Six- to eight-week-old female BALB/c mice were purchased from Andreevka Animal Center (Andreevka, Russia). Tumors were induced via subcutaneous injection of 1 × 10^6^ cells (4T1 and 4T1-Qt. Tumor size and animal weight were monitored twice a week. The tumor volume (V) was calculated as V = a2/2 × b, where a is the smaller of the two orthogonal sizes (a and b), measured by caliper. Animals were euthanized when the tumors reached 500 mm^3^ or body weight loss exceeded 10% with a lethal dose of isoflurane. For immunohistochemical analysis (12 days after 4T1 and 4T1-Qt cells implantation), isolated tumors were postfixed for 24 h in 4% paraformaldehyde solution in PBS, and 40–60 μm tumor slices were obtained using an HM 650v vibratome (Microm GmbH, Ettlingen, Germany). The tumor slices were stained by mono-clonal Antibody to DYKDDDDK Tag (L5) and Alexa Fluor 647 (1:100, BioLegend, San Diego, CA, USA), and nuclei were counterstained with DAPI (1:400, Sigma Aldrich, Darmstadt, Germany).

## 3. Results

### 3.1. Expression of Bacterial Genes in Eukaryotic Cells

We used two lentiviruses to obtain 4T1 cells with stable expression of *Q.thermotolerans* encapsulin-encoding genes: QtEnc^FLAG^-QtIMEF-encoding encapsulin protomer and cargo protein, and mZip14-encoding iron transporter. The expression of encapsulin genes was validated via RT-PCR (Figure 1a). The protein expression of QtEncFLAG protomer was confirmed using Western blot analysis against FLAG-tag on the protomer protein. Figure 1b demonstrates a band at approximately 35 kDa. The signal level in Western blot increased with the amount of cell lysate loaded into the gel.

We also performed direct immunofluorescence staining of 4T1-Qt cells using primary labeled monoclonal anti-DYKDDDDK Tag antibodies that bind to the FLAG sequence coexpressed with QtEncFLAG, i.e., on encapsulin protomer proteins. Wild-type 4T1 cells were used as a control. The high-intensity red fluorescent signal from Alexa Fluor 647 label in 4T1-Qt cells (Figure 2a) is visible in the micrograph. In control 4T1 cells stained with anti-DYKDDDDK Tag antibodies, a red fluorescent signal was not detected (Figure 2b).

Confocal microscopy resolution is not enough to detect individual 42 nm diameter encapsulins, so this imaging method is relatively coarse. In addition, we used the TEM to visualize the iron oxide nanoparticles formed inside the encapsulins, allowing us to estimate the efficiency of iron biomineralization and storage.

### 3.2. Iron Biomineralization Inside the Encapsulins

We hypothesized that when FAS is added to wild-type 4T1 cells, excess iron may have a toxic effect due to ferritin overload and accumulation of free iron in the cytoplasm. In contrast, in genetically modified 4T1-Qt, part of the iron ions will be sequestered into encapsulins, which may reduce toxicity. From the data presented in Figure 3, it can be seen that our hypothesis was confirmed in the 0.25–4 mM FAS concentration range.

Knowing the toxicity of FAS for 4T1 and 4T1-Qt cells, we decided to qualitatively assess iron accumulation in cells via Prussian blue staining using the same concentrations of FAS as in the cytotoxicity assay. In a series of micrographs, the light blue staining of iron deposits in 4T1-Qt cells after 24 h incubation with FAS in the 0.25–2 mM concentration range is detected (Figure 4a, stained areas are indicated by black arrows). At 4 mM FAS, deviations in cell morphology and a decrease in the number of cells per well are seen, consistent with the toxicity assay data (only 54% of 4T1-Qt cells remained viable after 24 h incubation with 4 mM FAS). In contrast, no staining of control 4T1 cells was detectable after 24 h incubation with FAS at the same concentrations (Figure 4b). We can also observe the toxic effect of FAS on 4T1 cells at 0.25–4 mM concentration, in the form of a significant decrease in the density of the cell monolayer, the presence of cell debris, rounded cells, and cells with atypical morphology.

AES was used to quantify the iron accumulation in 4T1 and 4T1-Qt cells after the incubation with 0.25 mM, 0.5 mM, 1 mM, and 2 mM FAS (Figure 5).

It was found that iron accumulates in 4T1-Qt cells in a dose-dependent manner, and intracellular iron concentration increases proportionally to the amount of FAS added to the growth medium. The iron accumulation in 4T1-Qt cells is significantly higher than the same value in wild-type 4T1 cells.

Finally, we performed TEM imaging of encapsulins in 4T1-Qt cells after the incubation with 2 mM FAS for 24 h. We expected to find encapsulin-derived iron oxide nanoparticles within each cell. However, during TEM micrographs analysis, we noticed that the number of nanoparticles per cell is quite variable. For example, some of the cells contained electron-dense particles (Figure 6a); however, there were also cells without nanoparticles inside the encapsulin shells (Figure 6b). Both vectors used for transduction contained the same antibiotic resistance genes (puromycin). Thus, there were not only cells where both transgenic sequences survived during the selection, but also cells without mZip14. For MRI studies, such heterogeneity is unacceptable; therefore, it was necessary to obtain a more homogeneous cell line in terms of nanoparticle content.

### 3.3. Genetically Encoded Labels for MRI

To select the cells with high numbers of iron oxide-loaded encapsulins, MACS of 4T1-Qt cells preincubated with 2 mM FAS was performed. The magnetic sorting efficiency was about 2% (in other words, 2% of target cells were retained within the magnetic column). The sorted 4T1-Qt cells were then expanded and used to determine the T2 relaxation time in vitro. The same was conducted for the sorted 4T1-Qt cells after 24 h of incubation with 1 mM and 2 mM FAS. Wild-type 4T1 cells were used as a control (Figure 7).

The relaxation time of wild-type 4T1 cells after their incubation with FAS was not measured. It is known that, in cells dying by ferroptosis, the intensity of the MR signal may increase [14,15], and this experiment was deemed inappropriate in our case (cf. FAS toxicity data presented above).

It is well-known that xenogeneic proteins expression can affect cell growth rate and their tumorigenic potential by suppression of the tumor growth via immune mechanisms [16]. For a comparative assessment of the growth dynamics of tumors obtained from 4T1 and 4T1-Qt cells, tumor volumes were measured on different days after subcutaneous tumor cells implantation. Transgenic and wild-type tumors transplanted into mice reached detectable size 3–5 days after implantation. Our experiments showed that the dynamics of 4T1 and 4T1-Qt tumor growth did not differ (Figure 8).

This data indirectly confirm the absence of 4T1-Qt cells immunogenicity in immunocompetent mice.

We also provide direct immunofluorescence staining of 4T1- and 4T1-Qt-cell-induced tumor sections, using primary labeled antibodies to FLAG Tag sequence. Figure 9 shows confocal images of the tumor sections obtained 12 days after the injection of 4T1 and 4T1-Qt cells. Confocal microscopy images demonstrate a bright red fluorescent signal in 4T1-Qt cells induced tumor section. In control 4T1-cell-induced tumors stained with anti-FLAG Tag antibodies, a red fluorescent signal was not detected.

## 4. Discussion

The choice of the 4T1 cell line was based on the growth and metastasis of these cells in female BALB/c mice, closely mimicking stage IV human breast cancer. Tumors induced by subcutaneous injection of 4T1 cells in mice spontaneously metastasize to the lungs in almost 100% of cases, as well as to bones, lymph nodes, and, less often, to the brain [17,18]. We hope that the encapsulin-based label will allow us to study the growth and metastasis of 4T1-Qt tumors in vivo via MRI. Our previous study [13] has shown that the stable heterologous expression of *Myxococcus xanthus* encapsulin genes can be achieved in human mesenchymal stem cells (MSCs). The presence of transgenic sequences in cells did not reduce the rate of cell proliferation, and the T2 relaxation time of MSCs containing nanoparticles in encapsulins was significantly lower than the T2 relaxation time of intact MSCs. Another work [19] demonstrated that the encapsulin cargo system from *Q. thermotolerans* may be suitable for HepG2 hepatocellular carcinoma cells vizualisation. Thus, in the future, encapsulin-based genetic labels might be successfully used for MRI monitoring of eukaryotic cells.

Compared to our previous work, the label has been optimized; namely, two viral vectors instead of three are used to obtain a cell line stably expressing encapsulin genes. In addition, the efficiency of lentiviral transduction of malignant cells was higher compared to stem cells. Another significant improvement of this study is MACS, which is well-suited for tumor cells despite their low sorting efficiency. Due to the rapid proliferation of malignant cells, it is still possible to obtain a sufficiently large encapsulin-enriched cell fraction. 

It is well-known that iron metabolism in tumor cells is often altered compared to non-malignant ones, which also applies to the 4T1 cell line. As in many carcinomas, 4T1 cells have an increased expression level of a type 1 transferrin receptor (TfR1) gene [20]. More importantly, these cells have a reduced expression of a gene encoding ferroportin (FPN) [21]—a transmembrane protein that transports iron ions from cells to the extracellular environment. As mentioned before, in *Q.thermotolerans*, encapsulins presumably perform a protective function by sequestering free cytoplasmic iron. Our data suggest that this protective mechanism can also be realized in eukaryotic cells with a heterologous expression of encapsulin genes.

AES data indicate that iron accumulation in 4T1-Qt cells is dose-dependent. The intracellular iron concentrations per cell for 4T1 and 4T1-Qt cells after the incubation with 2 mM FAS are almost an order of magnitude different (0.1 ± 0.05 pg/cell and 0.7 ± 0.2 pg/cell, respectively). For a 4T1-Qt line obtained in this study, the maximum concentration of FAS allowing us to reach the maximum concentration of intracellular iron while maintaining the cell viability is 2 mM. In wild-type 4T1 cells, iron is stored mostly in the protein ferritin. Each ferritin complex can accumulate about 4000 iron atoms, while in transgenic 4T1-Qt cells, iron storage also occurs inside the encapsulins, which can accumulate more than 30,000 iron atoms inside its shell. Thus, while intact cells die due to iron overload, 4T1-Qt cells remain viable because extra iron added in the cell culture medium is deposited inside encapsulins shells. That is why iron accumulation in 4T1-Qt cells is significantly higher than in wild-type 4T1 cells.

MRI data have shown lower T2 relaxation times for MACS-separated 4T1-Qt cells after 24 h incubation with FAS at 1 mM and 2 mM concentrations (120 ± 14 ms and 134 ± 15 ms, respectively) in comparison with 4T1 cells (294 ± 53 ms).

Finally, preliminary in vivo studies showed no significant differences in the dynamics of tumor growth, which indicates that the genetically encoded label does not affect cell proliferation in mouse breast carcinoma model, and the encapsulin encoding sequence is stably present in 4T1-Qt cells, even 12 days after implantation in mice.

## 5. Conclusions

In this work, we describe the 4T1 mouse carcinoma cell line stably expressing *Qt* encapsulin genes for the first time. The latter do not alter the viability and proliferation of 4T1-Qt cells; moreover, *Qt* encapsulins have a protective effect against high concentrations of iron ions, at the same time providing a dose-dependent iron accumulation in 4T1-Qt cells. Finally, in vitro MRI study showed a decrease in T2 relaxation time for magnetically sorted 4T1-Qt cells compared to wild-type 4T1 cells. We believe that this is an essential step towards the future in vivo MRI monitoring of malignant cells.

## Figures and Tables

**Figure 1 nanomaterials-12-01657-f001:**
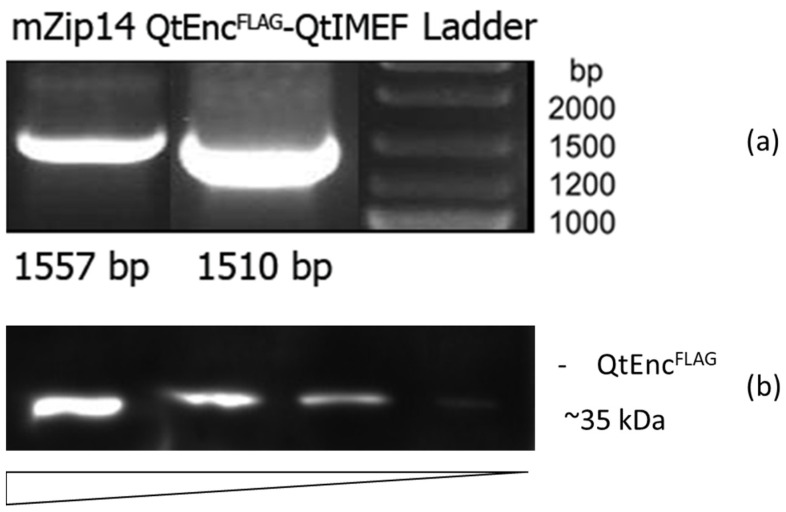
(**a**) RT-PCR analysis of 4T1-Qt cells; (**b**) Western blot analysis against FLAG-tag on *Q.thermotolerans* encapsulin protomer proteins in 4T1-Qt cells.

**Figure 2 nanomaterials-12-01657-f002:**
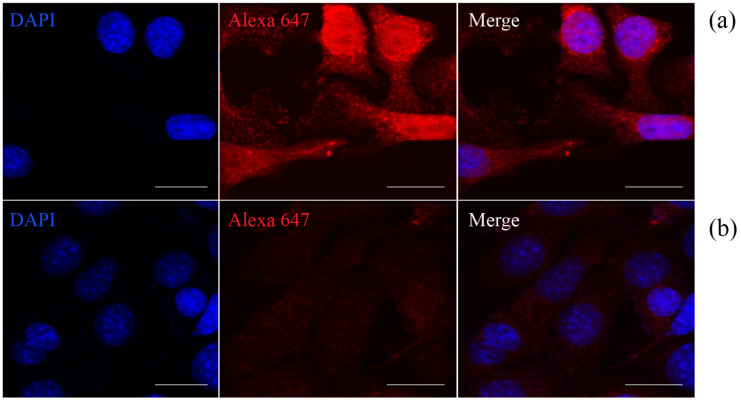
4T1-Qt (**a**) and 4T1 (**b**) cells stained with Alexa Fluor 647 anti-DYKDDDDK Tag antibody (red fluorescence). Nuclei were counterstained with DAPI (blue fluorescence). Laser scanning confocal microscopy, scale bar 20 μm.

**Figure 3 nanomaterials-12-01657-f003:**
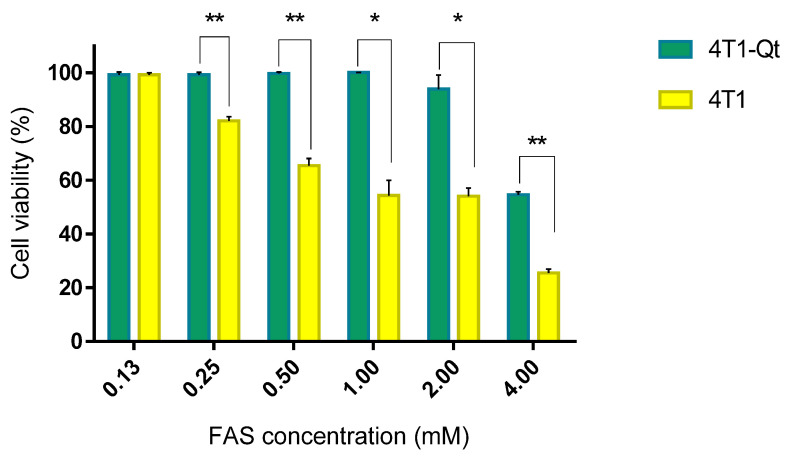
Cytotoxicity assay of various FAS concentrations in the growth medium for 4T1 and 4T1-Qt cells. The data are shown as the mean + S.D. of three independent experiments. *p* values were calculated using a one-tailed *t*-test, assuming unequal variances (** indicate *p*-value < 0.001, * indicate *p*-value < 0.05).

**Figure 4 nanomaterials-12-01657-f004:**
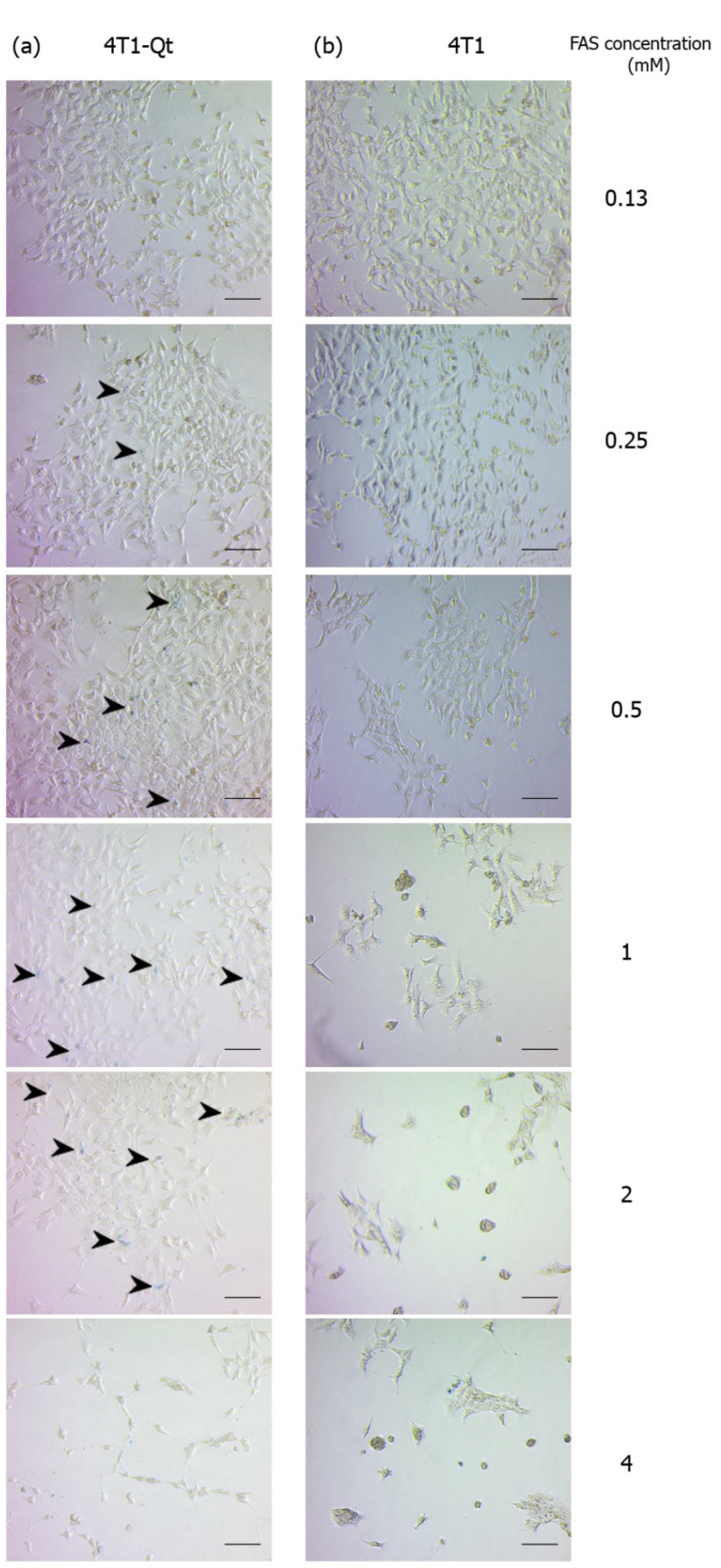
Prussian blue staining of 4T1-Qt (**a**) and 4T1 (**b**) cells after 24 h incubation with 0.13–4 mM FAS. Bright-field microscopy, scale bar 50 μm. Black arrows indicate iron deposits in 4T1-Qt cells.

**Figure 5 nanomaterials-12-01657-f005:**
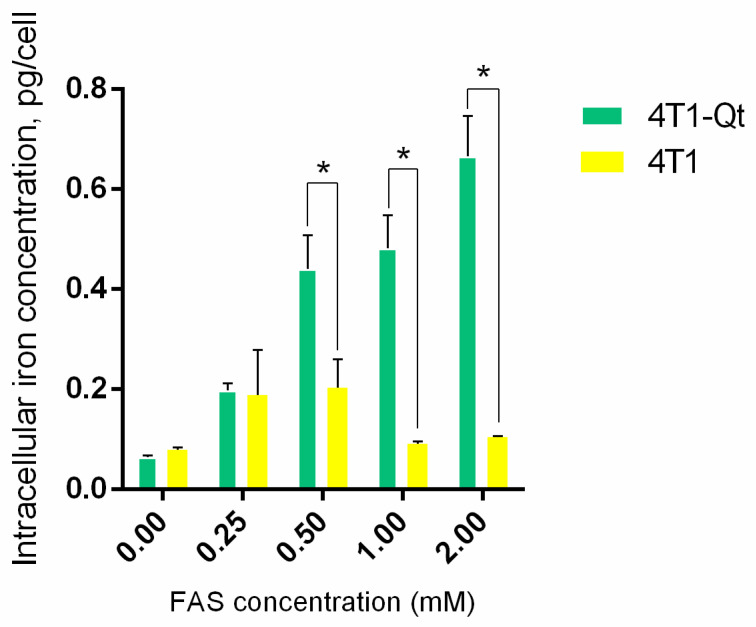
Cellular iron content in 4T1 and 4T1-Qt cells quantified by AES spectrometry. The data are shown as the mean + S.D of three independent experiments, *p* values were calculated using a one-tailed *t*-test, assuming unequal variances (* indicate *p*-value < 0.05).

**Figure 6 nanomaterials-12-01657-f006:**
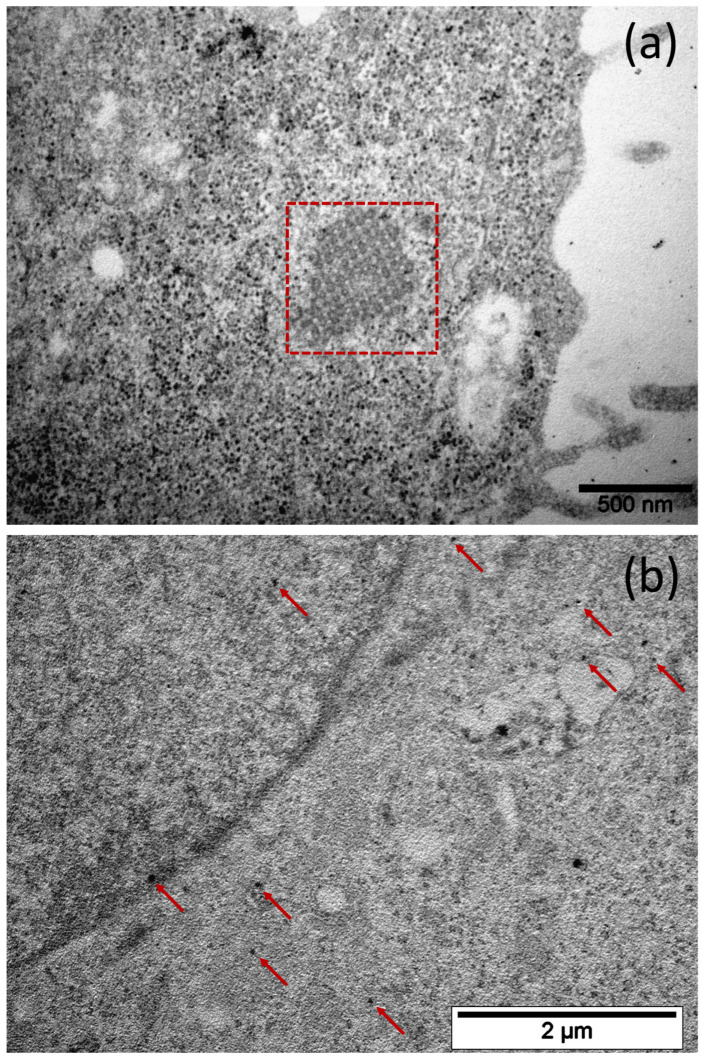
Bright-field TEM image of an ultrathin section of a 4T1-Qt cell, aggregated encapsulin shells in the cytoplasm are highlighted by a square (**a**), and red arrows indicate electron-dense nanoparticles in 4T1-Qt cell cytoplasm cell, scale bar 500 nm and 2 µm, respectively (**b**).

**Figure 7 nanomaterials-12-01657-f007:**
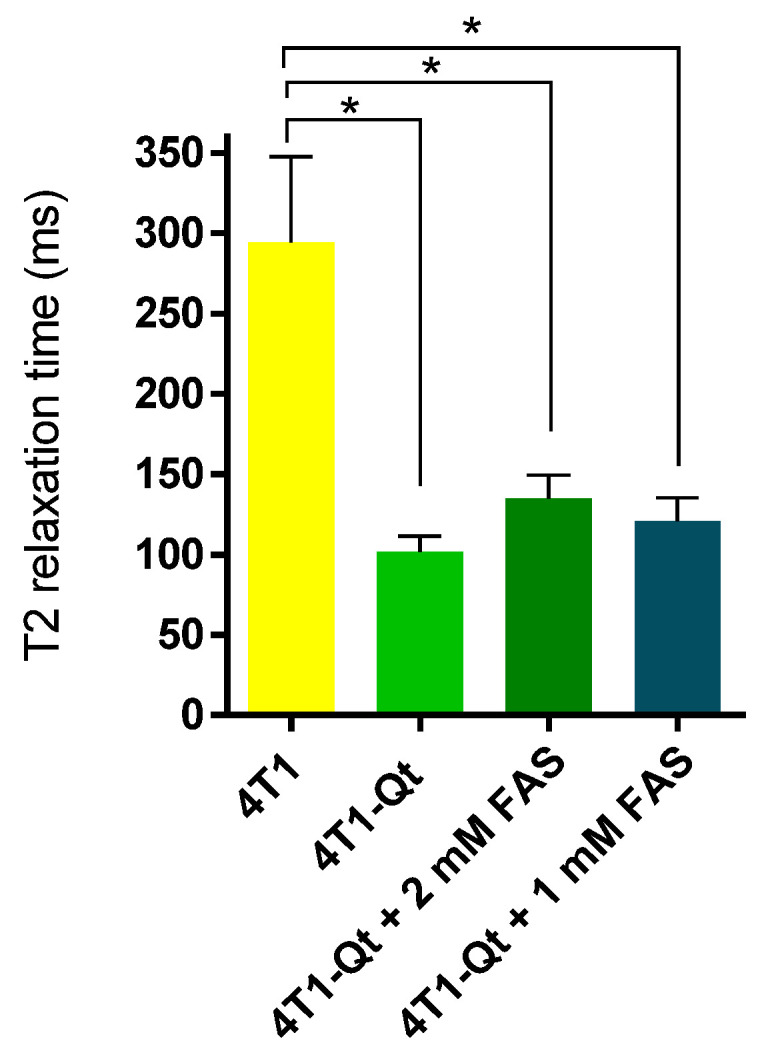
T2 relaxation time for 4T1 and 4T1-Qt cells. The data are shown as the mean + S.D. Statistical analysis was performed using an unpaired *t*-test (* corresponds to *p*-value < 0.05).

**Figure 8 nanomaterials-12-01657-f008:**
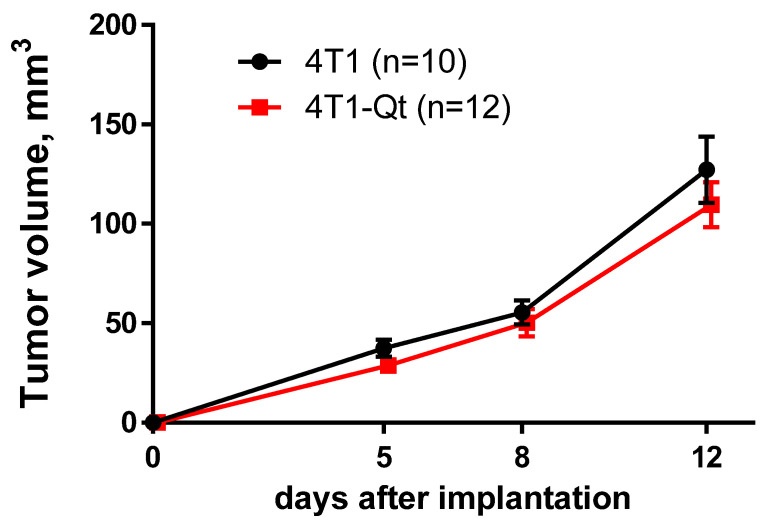
Tumor growth dynamics after subcutaneous implantation of wild-type 4T1 and 4T1-Qt cells. The data are shown as the mean + S.E.M.

**Figure 9 nanomaterials-12-01657-f009:**
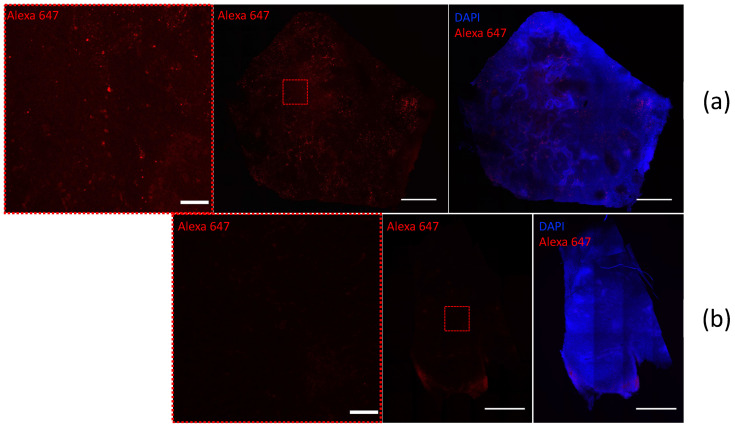
4T1-Qt- (**a**) and 4T1- (**b**) cell-induced tumors sections stained with Alexa Fluor 647 anti-DYKDDDDK Tag antibody (red fluorescence). Nuclei were counterstained with DAPI (blue fluorescence). Laser scanning confocal microscopy, scale bar 100 and 1000 μm.

## Data Availability

All important data are included in the manuscript.

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
