# Peer review of "New Approach to Non-Invasive Tumor Model Monitoring via Self-Assemble Iron Containing Protein Nanocompartments"

_nanomaterials, 2022, doi:10.3390/nano12101657_

Round 1

Reviewer 1 Report

The manuscript entitled “New approach to non-invasive tumor model monitoring via self-assemble iron containing protein nanocompartments”

The authors have presented a new genetically encoded label based on Quasibacillus thermotolerans encapsuling using in mouse 4T1 breast carcinoma cells. The iron oxide nanoparticles are formed by a ferroxidase oxidizes Fe2+ to Fe3+ which proving 4T1Qt cell tracking by MRI. Finally, in vitro MRI study showed a decrease in T2 relaxation time for magnetically sotred 4T1-Qt cells compared to wild-type 4T0 cells. However, few points are still not clear in this review. Therefore, I recommend the paper for the publication after minor modification.

  1. The authors have mentioned that the diameter of iron oxide nanoparticles is around 42 nm which is not easy to obtain the result in confocal microscopy as shown in Fig. 2. Therefore, the TEM is used to check the iron oxide nanoparticles is formed inside the encapsulins. However, we still can’t figure out this result from Fig. 6 TEM micrographs analysis. Besides, the authors should indicate where the iron oxide nanoparticle is in Fig. 6.
  2. The authors should give an more detail expression why the iron accumulation in 4T1-Qt cells is significantly higher than the same value in wild-type 4T1 cells.  

Author Response

We thank the Reviewer for these valuable comments.

  1. The presence of iron oxide nanoparticles was not detected in 4T1-Qt cell in Figure 6, this micrograph is an example of cells with only encapsulin shells, but without nanoparticles inside. However, we agree that one more illustration of 4T1-Qt cells with nanoparticles inside the encapsulins is necessary. We added another one TEM micrograph on Figure 6 (fig. 6b).
  2. We agree, the explanation was added to the manuscript.

Reviewer 2 Report

This work focuses on the development of an alternate approach to using an animal model for breast carcinoma to assess tumor growth, metastasis, and angiogenesis. The technique is based on extended visualization of the cancer cells in a non-invasive manner using a genetically encoded label based on Quasibacillus thermotolerans encapsulin expressed in mouse 4T1 breast carcinoma cells.The work could be a value-add if the system was applied preclinically for breat cancer diagnosis, biomarker targeting or demonstrating a treatment intervention to validate the performance of the new system. The novelty on this aspect is not clear and is critical before publication. Below are recommendations:

1) The language and grammar use needs to be improved drastically throughout the manuscript.

2) Have the authors explored confounding variables to iron homeostasis upon application of the system.

3) Studies on the activity (half-life) of the new system is required to understand its potential applicability.

4) Concentration-dependant data is needed on the encapsulin isolation and activity band for potential detection.

5) Magnetic field chracterization studies (e.g. SQUID) is needed to confirm the magnetic activity of the nanoparticles formed inside encapsulins.

Author Response

We thank the Reviewer for these helpful recommendations.

  1. We agree, corrections were done in the manuscript.
  2. This work was not done in this experimental work. Our work was mainly concentrated on proof of principle using genetically encoded encapsulin based system as an MRI label.
  3. For the genetically encoded label the term “half-life” in not fully correct. In the case of lentiviral gene transfer method the foreign DNA (in our work – Q.thermotolerans encapsulin encoding DNA) is integrated into the chromosomes and gets passed on to future generations of the cell, so the transgene becomes part of the host genome and is therefore replicated. In our study QtEncFLAG-QtIMEF and mZip14 expression in 4T1-Qt cells transduced with the lentiviral vectors persisted for more than 6 weeks. Also the encapsulin encoding sequence is stably present in 4T1-Qt cells even 12 days after implantation in mice.
  4. We are so sorry, but your comment not entirely clear to us. The aim of this work was to obtain a new cell line, which is often used as a model for measuring the effectiveness of therapy both in vitro and in vivo. Thanks to our genetically encoded construct, it is possible to observe the formation of new tumor foci (metastases) and / or a decrease in tumor volume due to any investigated factors (for example, the study of new chemotherapeutic drugs) via MRI. As shown by our in vivo experiments, this line did not differ in tumor growth rate compare wild-type 4T1. The isolation of encapsulins from bacteria and the study of their efficiency in the accumulation of iron ions and their role in the metabolism of bacterial strains (in this case, the kute strain) were studied earlier in publications. However, the isolation of encapsulins from eukaryotic cells and their study may be of scientific interest, but in this work we pursued other goals, and this article is a step towards the potential application of such interesting compartments as encapsulin to biomedicine. We are planning to continue this work in different directions at including isolation of encapsulins from eukaryotic cells.

In our work, we analyzed the cytotoxicity of FAS at various concentrations on two cell lines: 4T1 wild-type and 4T1-Qt (Figure 3 in the manuscript). Our results showed that at concentrations of 1 and 2 mM FAS, there is a dramatic difference in survival rate between two cell lines (4T1 and 4T1-Qt). At a higher concentration of FAS, the survival of the trasgenic line decreases, which does not allow us to use such an amount of iron in further studies. We also showed the accumulation of iron ions per cell (Figure 5), which shows a significant difference in the amount of iron between two cell lines (including concentrations of 1 and 2 mm FAS). Finally, we studied the relaxation time of the cell line we had obtained (when incubated with 1 and 2 mM FAS) and compared the obtained values with the relaxation time of wild-type 4T1. Our data showed a significant decrease in T2, which indicates the possible use of 4T1-Qt cells in long-term MRI monitoring.

  1. We agree with the Reviewer. Indeed, we have not studied magnetic properties; however, the fact that the resulting cells were isolated by magnetically activated cell sorting and the fact that the cells have a low T2-relaxation time indirectly confirm the presence of magnetic properties in the obtained cell line. Also overall amount of magnetic material do not allow to perform any direct magnetic measurements.